# Solidification Mechanism of Pb and Cd in S^2−^-Enriched Alkali-Activated Municipal Solid Waste Incineration Fly Ash

**DOI:** 10.3390/ma16103728

**Published:** 2023-05-14

**Authors:** Qi Xue, Yongsheng Ji, Zhanguo Ma, Zhongzhe Zhang, Zhishan Xu

**Affiliations:** 1State Key Laboratory for Geomechanics and Deep Underground Engineering, China University of Mining and Technology, Xuzhou 221116, China; xueqichn@cumt.edu.cn (Q.X.); zgma@cumt.edu.cn (Z.M.); 2Jiangsu Collaborative Innovation Center for Building Energy Saving and Construct Technology, Jiangsu Vocational Institute of Architectural Technology, Xuzhou 221116, China; zhangzzhaos@163.com; 3Jiangsu Key Laboratory Environmental Impact and Structural Safety in Engineering, China University of Mining and Technology, Xuzhou 221116, China; xuzhishan312@163.com

**Keywords:** MSWI fly ash, S^2−^-enriched modification, alkali-activated cementitious material, S^2−^-enriched alkali-activated slag (SEAAS), heavy metal solidification

## Abstract

S^2−^-enriched alkali-activator (SEAA) was prepared by modifying the alkali activator through Na_2_S. The effects of S^2−^-enriched alkali-activated slag (SEAAS) on the solidification performance of Pb and Cd in MSWI fly ash were investigated using SEAAS as the solidification material for MSWI fly ash. Combined with microscopic analysis through scanning electron microscopy (SEM), X-ray fluorescence spectroscopy (XRF), X-ray diffraction (XRD), and Fourier transform infrared spectroscopy (FT-IR), the effects of SEAAS on the micro-morphology and molecular composition of MSWI fly ash were studied. The solidification mechanism of Pb and Cd in S^2−^-enriched alkali-activated MSWI fly ash was discussed in detail. The results showed that the solidification performance for Pb and Cd in MSWI fly ash induced by SEAAS was significantly enhanced first and then improved gradually with the increase in dosage of ground granulated blast-furnace slag (GGBS). Under a low GGBS dosage of 25%, SEAAS could eliminate the problem of severely exceeding permitted Pb and Cd in MSWI fly ash, which compensated for the deficiency of alkali-activated slag (AAS) in terms of solidifying Cd in MSWI fly ash. The highly alkaline environment provided by SEAA promoted the massive dissolution of S^2−^ in the solvent, which endowed the SEAAS with a stronger ability to capture Cd. Pb and Cd in MSWI fly ash were efficiently solidified by SEAAS under the synergistic effects of sulfide precipitation and chemical bonding of polymerization products.

## 1. Introduction

The popularization and application of waste incineration technology have played a significant role in reducing urban solid waste [1,2], but not without producing some problems. By the end of 2020, the total amount of waste incineration for China was 216 million tons, and MSWI fly ash production was about 10 million tons. By 2025, Chinese MSWI fly ash production is expected to reach 13 million tons [3]. MSWI fly ash is enriched with high concentrations of Cd, Pb, Zn, Cu, and other harmful heavy metals that have the motential to migrate [4]. For heavy metal ions in MSWI fly ash, the leaching concentrations of Zn and Cu generally meet the limits of the existing hazardous waste standards, while those of Pb and Cd far exceed the limits required by these standards. As a result, MSWI fly ash in which Pb and Cd seriously exceed the standards is classified as hazardous waste. Therefore, Pb and Cd in MSWI fly ash must be solidified to meet the requirements specified in the standards before the fly ash can be transported landfills or used for subsequent resource applications [5].

There are three main methods for solidifying heavy metals in MSWI fly ash, including electrochemical extraction [6,7], solidification by cementitious materials [8,9,10], and melting vitrification [11,12,13]. Solidification by cementitious materials is considered the most suitable method for MSWI fly ash because of its simple treatment and sound economic and environmental effects [14]. In addition, the solidified body of fly ash after such solidification treatment has a desirable mechanical strength, which makes it easier to subsequently transport and dispose of in landfills [15].

Cement solidification technology is one of the most mature among the solidification technologies of cementitious materials [16,17]. In cement solidification technology, the main hydrated product, i.e., C-S-H gel, can physically adsorb and wrap the free heavy metals in fly ash, thus reducing the toxicity of MSWI fly ash [18,19]. However, if corrosive media or other external forces damage the solidified body, the heavy metals adsorbed by C-S-H gel will be dissolved in large quantities [20]. Hence, the solidification characteristics of cement mainly based on physical activity cannot achieve reliable solidification of free heavy metals in MSWI fly ash.

In addition to traditional cement solidification technology, alkali-activated slag (AAS), a new type of low-carbon and low-pollution cementitious material, also shows application prospects in heavy metal solidification [21]. AAS is one of the alkali-activated cementitious materials that forms three-dimensional network polymerization products by mineral polycondensation under alkaline conditions [22]. Because of its unique polymerization structure and high strength properties [23], it has apparent advantages in solidifying heavy metals [24]. El-Eswed BI et al. [25] and Pereira CF et al. [26] analyzed the heavy metal solidified body of alkali-activated cementitious material using SEM and FT-IR. The results showed that the addition of Pb, Cu, Cd, and Cr did not change the original structure of the silicon−oxygen tetrahedron and the aluminum−oxygen tetrahedron of the alkali-activated cementitious material, but participated in the formation of the polymerization reaction structure by balancing the charge, thus being effectively solidified in the skeleton system of the polymerization product by chemical bonding.

Although AAS, an alkali-activated cementitious material, has more advantages than cement when solidifying heavy metals in MSWI fly ash, the different ionic radii and coordination bond characteristics of various heavy metal ions result in the different abilities of AAS in terms of solidifying the heavy metal ions in MSWI fly ash through chemical bonding. Xu et al. [27] used alkali-activated cementitious material to solidify heavy metals. The results showed that alkali-activated cementitious material had the best solidification effect on Pb and Ni, but had a poor solidification effect on Zn, Cu, and Cd. Jaarsveld et al. [28] found that the solidification effect of Pb in alkali-activated cementitious materials was better than that of Cu. According to the BET test results, the ionic radius of Pb was larger than that of Cu, so the chemical bonding degree of Pb was better than that of Cu.

Explorative studies by the authors stated that AAS could achieve high-efficiency solidification for most heavy metal ions, such as Pb, in MSWI fly ash. The solidification rate of these heavy metal ions can reach more than 90% under the premise of a low GGBS dosage. However, for Cd and Se heavy metal ions, there is a problem of poor solidification performance by AAS, and the GGBS dosage is generally more than 55%. According to the requirements of the existing hazardous waste standards, it can be considered that the solidification of heavy metals in MSWI fly ash follows the “barrel effect”. Therefore, it is of great research significance to compensate for the poor solidification performance of some heavy metal ions in MSWI fly ash by AAS to realize the high-efficiency solidification of all heavy metal ions in MSWI fly ash by AAS.

Na_2_S is a type of inorganic chemical agent commonly used to solidify heavy metal ions in soil or sewage [29]. It can react with heavy metal ions to form stable sulfide precipitates, thus effectively solidifying heavy metal ions [30]. However, the affinity of S^2−^ in Na_2_S to various heavy metal ions, hence, its solidification effect on different heavy metals, is also selective. Moreover, applying Na_2_S requires a hydrolysis reaction to precipitate S^2−^; however, nearly half of Na_2_S will react with water to produce H_2_S gas when Na_2_S is dissolved in a traditional neutral solution, which not only pollutes the atmosphere, but also loses S^2−^. The modification of traditional alkali activators by Na_2_S demonstrates the potential application value, as, fortunately, no H_2_S is created when Na_2_S is dissolved in a highly alkaline solution.

The purpose of this paper was to solve the problem that Pb and Cd in MSWI fly ash severely exceed the standard, and to compensate for the poor solidification efficiency of AAS on some heavy metal ions in MSWI fly ash. In the current study, the S^2−^-enriched alkali-activator (SEAA) was prepared by modifying the alkali activator with Na_2_S. SEAA and S^2−^-enriched alkali-activated slag (SEAAS) were used as solidification materials to solidify Pb and Cd in MSWI fly ash, respectively. The optimal dosage of Na_2_S in SEAAS was determined by analyzing the changes in the leaching concentrations of Pb and Cd in the leachate before and after the solidification treatment of MSWI fly ash, and the effects of GGBS dosage in SEAAS on the solidification performance for Pb and Cd in the fly ash were investigated. By measuring the PH value of the activator and the compressive strength of the solidified bodies, the effect of S^2−^-enriched modification on the alkalinity of the activator and the mechanical properties of the solidified bodies was studied. Combined with SEM, XRF, XRD, and FT-IR microscopic analysis methods, the micro-morphology and molecular composition of the SEAAS solidified body were determined. Based on the research results, the solidification mechanism of Pb and Cd in S^2−^-enriched alkali-activated MSWI fly ash was revealed.

## 2. Materials and Methods

### 2.1. Materials

(1)Ground granulated blast-furnace slag (GGBS) powder

GGBS produced by a local cement factory in Xuzhou, China, was used. The specific surface area of the GGBS powder was 416 m^2^/kg and the density was 2700 kg/m^3^. Table 1 shows the chemical composition of GGBS powder. According to the chemical composition of GGBS powder, the basicity coefficient and mass coefficient were calculated as 0.88 and 1.96, respectively. The particle size was measured using a laser particle analyzer in the range of 0.1~73 μm with an average particle size of 12.76 μm.

(2)MSWI fly ash

MSWI fly ash from a waste incineration plant in Yangzhou, China, was used. The appearance of the MSWI fly ash is shown in Figure 1. It can be seen from the figure that the MSWI fly ash was a light yellow powder. The MSWI fly ash was dechlorinated fly ash with a specific surface area of 314 m^2^/kg and a density of 2342 kg/m^3^. Table 1 shows the main chemical components of the MSWI fly ash. It can be seen from Table 1 that the main chemical components of the fly ash were SiO_2_, A1_2_O_3_, and CaO.

According to HJ/T 300-2007 [31], the leaching concentrations of heavy metal ions in the MSWI fly ash leaching solution were analyzed using an ICP-MS laser denudation-plasma mass spectrometer (Table 2). The Cd and Pb contents in the MSWI fly ash seriously exceeded the standard level by taking the concentration limits of leaching liquid pollutants in GB 5085.3-2007 [32] as the standard level, and the corresponding multiples of exceeding the standard were 8.33 times and 18.29 times, respectively.

(3)Other

Table 3 shows the specific parameters of the chemical reagents. Sodium silicate and sodium hydroxide were used as raw materials to prepare the alkali activator. The Na_2_O content in sodium silicate was 9.65%, the content of SiO_2_ was 25.22%, and the water content was 65%. Sodium sulfide(Na_2_S) and glacial acetic acid were used as raw materials to prepare the SEAA and heavy metal leaching agents, respectively.

### 2.2. Preparation of SEAA

SEAA was prepared by adding Na_2_S into the alkali activator. Sodium silicate and sodium hydroxide were prepared into liquid sodium silicate with a silicate modulus (SiO_2_/Na_2_O) of 1.5. The alkali activator was dissolved in deionized water, and Na_2_S was added while stirring. After the Na_2_S was completely dissolved, the SEAA was prepared.

### 2.3. Preparation of MSWI Fly Ash Solidified Body

The solidified body of MSWI fly ash was prepared to determine the effects of the solidification material on various properties of MSWI fly ash. The solidification material and MSWI fly ash were mixed according to the corresponding mix proportion (the specific mix proportion will be described separately in Section 2.4). The mixture was stirred until a liquid, viscous slurry was formed. The freshly mixed paste was injected into a test mold of 4 × 4 × 4 cm for compaction and forming, and then the mold was cured under the conditions at a temperature of 20 ± 3 °C and relative humidity of 95% or more. When the corresponding curing age was reached, the solidified body was taken out for use.

### 2.4. Experiment and Research Contents

#### 2.4.1. Effects of Na_2_S Dosage on the Solidification Performance for Pb and Cd by SEAA

(1)Mixing proportion and preparation of specimens

The SEAA solidified bodies with different Na_2_S dosages at a water−binder ratio of 0.4 were prepared. In different groups, the mass of fly ash was constant, and the dosages of Na_2_S were 0%, 0.5%, 1%, and 2% the mass of the fly ash. The alkali activator’s solid content (mass sum of SiO_2_ and Na_2_O) was fixed at 17.5% of the fly ash mass.

The preparation process of the SEAA solidified body is shown in Figure 2. SEAA was prepared according to Section 2.2, and then the SEAA solidified bodies were prepared according to Section 2.3 by mixing the SEAA with MSWI fly ash. By taking the Na_2_S solidified bodies with the same water−binder ratio and the same Na_2_S dosage as the control group, respectively, the effects of Na_2_S dosage on the solidification performance for Pb and Cd by SEAA were investigated.

(2)Preparation of heavy metal leaching agent

A solidified body cured for seven days was dried for 24 h at 65 °C, then crushed into disintegrating slag of the solidified body with a particle size ≤ 0.15 mm. According to HJ/T300-2007, the disintegrating slag of solidified body (20.0g) was weighed and put into a polyethylene bottle containing 0.3 mol/L glacial acetic acid diluent (400 mL), and the bottle was turned over and set at 30 r/min for 18 h at 23 ± 2 °C. After standing for 6 h, 10 mL of solution was extracted. The extract was filtrated through a mesh sieve of 0.8 μm to obtain the heavy metal leaching solution of the MSWI fly ash solidified body.

(3)Determination of heavy metal contents

An ICP-MS laser denudation plasma mass spectrometer (Agilent Technologies, NWR 213—7900, Santa Clara, CA, USA) was used to analyze the solidification performance of Pb and Cd in MSWI fly ash of the solidification materials. The solidification rate of a certain heavy metal ion in the solidified body was calculated according to Equation (1).
(1)$$\mathrm{Solidification}\;\mathrm{rate}=\frac{\rho_{0}-\rho_{1}}{\rho_{1}}\times 100\%$$
where $\rho_{0}$ is the leaching concentration of a certain heavy metal ion from MSWI fly ash before solidification, ppb, and $\rho_{1}$ depicts the leaching concentration of a certain heavy metal ion from MSWI fly ash after solidification, ppb.

#### 2.4.2. Effects of GGBS Dosage on the Solidification Performance for Pb and Cd by SEAAS

(1)Mixing proportion and preparation of specimens

SEAA with an optimal Na_2_S dosage was used to prepare the SEAAS solidified body with different GGBS dosages at a constant water−binder ratio of 0.4. In each group, the sum of the mass of MSWI fly ash and GGBS was constant. The doping method of GGBS was internal doping, and the dosage was 0%, 5%, 15%, 25%, 35%, 45%, 55%, and 65%, respectively. The solid content of the alkali activator was fixed at 17.5% of the mass of the powder.

The preparation process of the SEAAS solidified body was shown in Figure 3. The SEAA and MSWI fly ash were evenly mixed to prepare the SEAA solidified paste. Then, the fresh SEAA solidified paste was mixed with GGBS to prepare the SEAAS solidified body according to Section 2.3. The effects of GGBS dosage on the solidification performance for Pb and Cd by SEAAS were examined using the AAS solidified bodies with the same water−binder ratio and the same GGBS dosage as the control group, respectively.

(2)Preparation of the heavy metal leaching agent

The preparation of the heavy metal leaching agent was consistent with (2) in Section 2.4.1.

(3)Determination of heavy metal contents

The determination method of heavy metal contents was the same as (3) in Section 2.4.1.

#### 2.4.3. Effects of S^2−^-Enriched Modification on the Alkalinity of the Activator

(1)Preparation of testing fluid

The SEAA with the best Na_2_S content was selected, and the alkali activator and SEAA were prepared according to Section 2.2 to study the effects of S^2−^-enriched modification on the alkalinity of the activator.

(2)Determination of pH value

The pH value of the activator was determined by an acidometer (Ritz, PHB-5, Shanghai, China). The pH range of the acidometer was −2~18, and the accuracy was 0.01.

#### 2.4.4. Effects of S^2−^-Enriched Modification on the Mechanical Properties of Fly Ash Solidified Bodies

(1)Mixing proportion and preparation of specimens

The solidified bodies of SEAA with the optimal Na_2_S dosage and SEAAS with the optimal GGBS dosage were prepared according to Section 2.3. The AAS solidified body with the same GGBS dosage and the Na_2_S solidified body with the same Na_2_S dosage were also prepared in the same way as the control group to investigate the effects of S^2−^-enrich modification on the mechanical properties of the solidified bodies.

(2)Compressive strength test

The compressive strengths of the solidified bodies with curing ages of seven days and twenty-eight days were determined by an electronic universal testing machine following GB/T17671-2021 [33].

#### 2.4.5. Effects of SEAAS on the Micro-Morphology of MSWI Fly Ash

(1)Mixing proportion and preparation of samples

SEAA with the optimal Na_2_S dosage was selected to prepare a solidified body of SEAAS with an optimal GGBS dosage. The AAS solidified body with the same GGBS dosage was prepared as a reference to study the effects of SEAAS on the microstructure of MSWI fly ash.

After solidification treatment for seven days, the solidified body was split into blocks, and square blocks with a length and thickness of about 6 mm were screened. The blocks were dried at 65 °C for 24 h, then placed in an ion sputtering instrument for surface gold spraying to prepare the samples for SEM observation.

(2)SEM analysis

An FEI Quanta 250 environmental scanning electron microscope (SEM) was used to observe the micro-morphology of the samples. The samples for SEM observation were fixed on the sample stage, and the electron microscope parameters were adjusted to observe the samples in the acceleration voltage range of 5~10 kV and the probe current range of 1 pA~2 mA.

#### 2.4.6. Effects of SEAAS on the Chemical and Molecular Composition of MSWI Fly Ash

(1)Mixing proportion and preparation of samples

The SEEAS solidified body with the same ratio as Section 2.4.5 was prepared. The original MSWI fly ash and the AAS solidified body with the same GGBS dosage were used as a reference to study the effects of SEAAS on the chemical and molecular composition of MSWI fly ash.

The original MSWI fly ash and the solidified body with curing ages of seven days were dried at 65 °C for 24 h. They were then ground and sieved through a 325-mesh sieve, and the sieved powder was used as test samples for the XRF and XRD analysis. The sieved powder was mixed with KBr at a ratio of 1:8 and then pressed into flat specimens of 1 to 10 μm as FT-IR test samples using a hydraulic pelletizer with a 5-ton load capacity.

(2)XRF analysis

An X-ray fluorescence spectrometer (Bruker, S8 TIGER, Bremen, Germany) was used to determine the chemical composition and elemental content of the test samples.

(3)XRD analysis

The mineral composition of the test samples was analyzed using an X-ray diffractometer (Bruker, D8 Advance, Bremen, Germany) with diffraction angles ranging from 5° to 65° (2θ) in steps of 0.020° at a rate of 2°/min, and Cu–Ka radiation was applied.

(4)FT-IR analysis

The vibration patterns and molecular structures of the test samples were analyzed using an infrared spectrograph (Bruker, VERTEX 80v, Bremen, Germany). Each sample was scanned 32 times with a resolution of 2 cm^−1^, and the wavelength range was between 400 and 4000 cm^−1^.

## 3. Results and Discussion

### 3.1. Effects of Na_2_S Dosage on the Solidification Performance of SEAA for Pb and Cd

(1)Solidification performance for Pb

Figure 4a shows the effects of Na_2_S dosage on the solidification performance of SEAA for Pb. According to the findings, the original leaching concentration of Pb in MSWI fly ash was noted as 4572.78 ppb. When the Na_2_S dosages were 0.5%, 1%, and 2%, the leaching concentrations of the Na_2_S solidified body (control group) were determined as 3153.72, 1673.25, and 962.52 ppb, respectively, and the corresponding solidification rates were 31.03%, 63.44%, and 78.95%, respectively.

When the Na_2_S dosage was 0.5%, 1%, and 2%, the SEAA solidified body leaching concentrations were 2574.08, 1047.56, and 638.96 ppb, respectively, and the corresponding solidification rates were 43.71%, 77.09%, and 86.02%, respectively. The solidification rates of the SEAA solidified body increased by 12.68%, 13.68%, and 7.07% compared with those of the Na_2_S solidified body at each dosage level, respectively. These results showed that under the same Na_2_S dosage, the leaching concentration of Pb in the SEAA solidified body was slightly lower than that in the control group, indicating that the solidification performance of SEAA for Pb in MSWI fly ash was slightly better than the Na_2_S solidified body.

There was a negative correlation between Na_2_S dosage and Pb leaching concentration of the SEAA solidified body, especially in the range of 0% to 1% Na_2_S dosage, indicating that the increase in Na_2_S dosage could reduce the Pb leaching concentration of the SEAA solidified body. However, when the Na_2_S dosage was 2%, the leaching concentration of Pb in the SEAA solidified body was still higher than the standard limit of 250 ppb required by GB 5085.3-2007.

(2)Solidification performance for Cd

Figure 4b shows the effects of Na_2_S dosage on the solidification performance of SEAA for Cd. The original leaching concentration of Cd in MSWI fly ash was found to be 1250.77 ppb. When the Na_2_S dosages were 0.5%, 1%, and 2%, the leaching concentrations of the Na_2_S solidified body (control group) were 436.17, 115.13, and 36.01 ppb, respectively, and the corresponding solidification rates were 65.13%, 90.80%, and 97.12%, respectively.

When the Na_2_S dosages were 0.5%, 1%, and 2%, the leaching concentrations of SEAA solidified body was 137.35, 39.62, and 22.41 ppb, respectively, and the corresponding solidification rates were 89.02%, 96.83%, and 98.21%, respectively. The solidification rates of the SEAA solidified body increased by 23.89%, 6.04%, and 1.09% compared with those of the Na_2_S solidified body at each Na_2_S dosage level. These results indicate that the solidification performance of SEAA for Cd in MSWI fly ash was much better than the control group.

There was a stronger negative correlation between the Na_2_S dosage and Cd leaching concentration of the SEAA solidified body, especially in the range of the 0% to 0.5% Na_2_S dosage, indicating that the increase in Na_2_S dosage could significantly reduce the Cd leaching concentration of the SEAA solidified body. When the Na_2_S dosage was 0.5%, the leaching concentration of Cd in the SEAA solidified body was comparable to that in the research group with a Na_2_S dosage of 1%, and was lower than the standard limit of 150 ppb required by GB 5085.3-2007.

Figure 4a,b reveals that the leaching concentrations of Pb and Cd from the SEAA solidified body decreased significantly first and then slightly with the increase in Na_2_S dosage. In other words, the solidification performance of SEAA for Pb and Cd in MSWI fly ash was enhanced significantly first and then slightly with the increase in Na_2_S dosage. Quina MJ et al. used Na_2_S to solidify heavy metals in MSWI fly ash, and the results indicated that when the Na_2_S dosage was 3~10% of the weight of the MSWI fly ash, it could show a significant solidification performance [30]. The critical values where SEAA solidified body leaching concentrations of Pb and Cd were significantly reduced corresponding to the Na_2_S dosage of 1%, which indicated that the solidification performance of SEAA was more excellent. Therefore, the with a Na_2_S dosage of 1% was selected to prepare the SEAAS solidified body.

To sum up, within a low Na_2_S dosage range of 0.5~1%, although SEAA could effectively solidify Cd in MSWI fly ash and solved the problem that Cd seriously exceeded the standard limit, it could not effectively solidify Pb; thus, the leaching concentration of Pb from the SEAA solidified body was still higher than the standard limit. Therefore, it was difficult for SEAA to solve the problem that Pb in MSWI fly ash seriously exceeded the standard limit.

### 3.2. Effects of GGBS Dosage on the Solidification Performance of SEAAS for Pb and Cd

(1)Solidification performance for Pb

Figure 5a presents the effects of GGBS dosage on the solidification performance of SEAAS for Pb. As can be seen, when the GGBS dosage was 5%, 15%, and 25%, the leaching concentrations of Pb from the AAS solidified body (control group) were 1843.65, 712.45, and 232.13 ppb, respectively, and the corresponding solidification rates were 59.68%, 84.42%, and 94.92%, respectively. With the further increase in GGBS dosage to 35%, 45%, and 55%, the leaching concentration of Pb from the AAS solidified body decreased further, but tended to be gradual. When the GGBS dosage was 55%, the Pb leaching concentration of the AAS solidified body was 21.32 ppb, and the solidification rate was only 3.12% higher than that of the AAS solidified body with a GGBS dosage of 25%.

When the GGBS dosages were 5%, 15%, and 25%, the leaching concentrations of Pb from the SEAAS solidified body was 974.24, 357.15, and 91.48 ppb, respectively. The corresponding solidification rates were 80.88%, 90.01%, and 97.99%, respectively, which increased by 30.78%, 20.15%, and 13.06% compared with those of the AAS solidified body at each dosage level. These results indicate that SEAAS was slightly better than AAS for the Pb solidification of MSWI fly ash.

When the GGBS dosage was in the range of 0~25%, there was a strong negative correlation between GGBS dosage and the Pb leaching concentration of the SEAAS solidified body, indicating that the increase in GGBS dosage could significantly reduce the Pb leaching concentration of the SEAA solidified body, so that the leaching concentration of the SEAAS solidified body with 25% GGBS dosage was lower than the standard limit of 250 ppb required by GB 5085.3-2007.

With the further increase in GGBS dosage to 35%, 45% and 55%, there was a weak negative correlation between the GGBS dosage and Pb leaching concentration of the SEAAS solidified body, indicating that the Pb leaching concentration of the SEAAS solidified body was not significantly reduced in these GGBS dosage ranges.

(2)Solidification performance for Cd

Figure 5b shows the effects of GGBS dosage on the Cd solidification performance of SEAAS. As can be seen, when the GGBS dosages were 5%, 15%, and 25%, the leaching concentration of the AAS solidified body (control group) showed an obvious decreasing trend; however, the leaching concentration of the AAS solidified body with a GGBS dosage of 25% was 355.41 ppb, which was still higher than the standard limit of 150 ppb required by specifications. With the further increase in GGBS dosage to 35%, 45%, and 55%, the leaching concentration of the AAS solidified body was further reduced, but tended to be moderate. When the GGBS dosage was 55%, the Cd leaching concentration of the AAS solidified body was 123.9 ppb, lower than the standard limit of 150 ppb required by the specifications. Hence, AAS had a poor solidification performance for Cd in MSWI fly ash.

When the GGBS dosage was in the range of 0~15%, there was a strong negative correlation between the GGBS dosage and the Cd leaching concentration of the SEAAS solidified body. The Cd leaching concentration of the SEAAS solidified body with a 5% GGBS dosage was 127.13 ppb, and the corresponding solidification rate was 90.00%. This leaching concentration was equivalent to the AAS solidified body with a GGBS dosage of 55% (123.9 ppb), lower than the standard limit of 150 ppb required by GB 5085.3-2007, revealing that SEAAS had excellent solidification performance for Cd in MSWI fly ash.

With the further increase in GGBS dosage to 25%, 35%, 45%, and 55%, there was a weak negative correlation between the GGBS dosage and Cd leaching concentration of the SEAAS solidified body, indicating that the Cd leaching concentration of the SEAAS solidified body was not significantly reduced in these GGBS dosage ranges.

From Figure 5a,b, it can be seen that the leaching concentrations of Pb and Cd for the SEAAS solidified body decreased significantly first and then became steady with the increase in GGBS dosage, indicating that the solidification performance of SEAAS for Pb and Cd in MSWI fly ash was enhanced significantly first and then slightyly with the increase in GGBS dosage. This regularity was consistent with the research results of Zhang et al. on the solidification of waste incineration fly ash by AAS, which showed that the heavy metal leaching concentration of the AAS solidified body tended to decrease decreased significantly first and then slightly with the increase in GGBS dosage, and the GGBS dosage corresponding to the solidification of heavy metals to meet the standard was 45% [34]. The GGBS dosage of SEAAS for solidifying heavy metals to meet the standard was 25%, indicating that the solidification performance of SEAAS was better.

To sum up, in a low GGBS dosage range (0~25%), SEAAS can effectively solidify Pb and Cd in MSWI fly ash. It had a particularly excellent solidification performance for Cd, which successfully compensated for the poor solidification performance of traditional AAS for Cd in MSWI fly ash. Hence, with a low GGBS dosage of 25%, SEAAS could solve the problem that the contents of Pb and Cd in MSWI fly ash seriously exceeded the standard limits.

### 3.3. Effects of S^2−^-Enriched Modification on the Alkalinity of the Activator

Table 4 shows the pH values of the alkali activator and SEAA (1% Na_2_S dosage). As can be seen, the pH value of the alkali activator was 13.81, and the pH value of SEAA was 13.94. Compared with the alkali activator, the concentration of OH^−^ in SEAA increased by 0.225 mol/L.

The reason for the higher OH^−^ concentration of SEAA was that Na_2_S dissolved in water could form sodium hydroxide, as shown in Equation (2). Therefore, compared with the AAS activator with the same sodium silicate modulus, the concentration of OH^−^ in SEAA was increased, which provided a higher alkaline environment for the dissolution of S^2−^ and the polymerization reaction of powders (GGBS and MSWI fly ash).
Na_2_S + H_2_O = NaHS + NaOH(2)

### 3.4. Effects of S^2−^-Enriched Modification on the Mechanical Properties of MSWI Fly Ash Solidified Body

Figure 6 shows the compressive strength of different fly ash solidified bodies. As can be seen, after curing for seven days or twenty-eight days, the order of the compressive strength for the four types of solidified bodies was as the SEAAS solidified body (1% Na_2_S dosage + 25% GGBS dosage) > AAS solidified body (25% GGBS dosage) > SEAA solidified body (1% Na_2_S dosage) > Na_2_S solidified body (1% Na_2_S dosage).

The compressive strength of the Na_2_S solidified body was found to be significantly lower than that of other solidified bodies, indicating that the Na_2_S solution had a limited ability to activate the activity of MSWI fly ash. Compared with the Na_2_S solidified body, the seven day and twenty-eight day compressive strengths of the SEAA solidified and cured body increased by 112.59% and 78.07%, respectively, indicating that SEAA could activate the active components in MSWI fly ash well, thus improving the mechanical properties of the solidified body.

The compressive strength of the SEAAS solidified body was higher than that of the AAS-solidified body, and the former’s seven days and twenty-eight days compressive strengths increased by 10.41% and 11.21%, respectively, compared with those of the latter. These indicate that S^2−^ enrichment had no negative effect on the mechanical properties of the fly ash solidified body, but improved its mechanical properties. In addition, compared with those of the SEAA solidified body, the compressive strength of either the SEAAS solidified body or AAS solidified body increased by more than two times, indicating that the dosage of GGBS significantly improved the mechanical properties of the MSWI fly ash solidified body.

### 3.5. Effects of SEAAS on the Micro-Morphology of MSWI Fly Ash

Figure 7 presents the micro-morphology of the solidified bodies. Figure 7a illustrates the micro-morphology of the AAS solidified body (25% GGBS dosage) and Figure 7b shows the micro-morphology of the SEAAS solidified body (1% Na_2_S dosage + 25% GGBS dosage).

(1)Micro-morphology of AAS solidified body

It can be seen from Figure 7a that some irregular particles were scattered on the surface of the AAS solidified body, which indicates that a small amount of MSWI fly ash and GGBS particles in the AAS solidified body did not fully participate in the polymerization reaction. At the same time, there were also a large number of main cracks (1~2 μm) and secondary cracks (0.5~1 μm) in the micro-morphology of the sample. The main cracks were crisscrossed and irregularly extended everywhere, affecting the solidified body’s integrity. The end of the secondary cracks continued to expand and extend, and its extension direction was in the same direction as the main cracks, which had not only a negative effect on the mechanical properties of the solidified body, but also weakened the physical barrier effect of AAS on some free heavy metals in MSWI fly ash.

(2)Micro-morphology of SEAAS solidified body

As can be seen from Figure 7b, compared with the AAS solidified body, most of the particles on the surface of the SEAAS solidified body disappeared, and the sample showed a uniform and dense gel. At the same time, there were no main cracks and secondary cracks in the micro-morphology of the sample, only a small amount of microcracks (<0.5 μm), and its extension and penetration were significantly reduced. Combined with the results of Section 3.3, SEAA had a higher OH^−^ concentration, which made the polymerization reaction between GGBS and SEAA more complete, thus producing a more compact gel structure. Therefore, the SEAAS solidified body had better mechanical properties and a more prominent physical barrier effect on free heavy metals.

### 3.6. Effects of SEAAS on the Chemical and Molecular Composition of MSWI Fly Ash

#### 3.6.1. XRF Analysis

Table 5 shows the XRF of MSWI fly ash and solidified bodies. The main components of MSWI fly ash were CaO, SiO_2_, Al_2_O_3_, etc. Pb and Cd were present in MSWI fly ash in the form of most oxides and a small portion of non-oxides. Compared with the MSWI fly ash, the content of CaO, SiO_2_, Al_2_O_3_, and Na_2_O in the AAS solidified body (25% GGBS dosage) increased, and the content of Pb, Cd, PbO, and CdO decreased.

Compared with the AAS solidified sample, the contents of CaO, SiO_2_, and Al_2_O_3_ in the SEAAS solidified body (1% Na_2_S dosage + 25% GGBS dosage) were approximately the same. The difference was that its Na_2_O content increased, and the content of sulfur increased significantly, indicating that S^2−^-enriched modification increased the alkali content and sulfur content in the solidified body. In addition, the increase in sulfur content in the SEAAS solidified body was accompanied by an increase in the content of Cd and Pb due to the formation of sulfide precipitates. The increase in Cd was greater, indicating that SEAAS has a stronger ability to capture Cd to form stable sulfides.

#### 3.6.2. XRD Analysis

The XRD of MSWI fly ash and solidified bodies is shown in Figure 8. It can be seen from the figure that the main diffraction peaks of MSWI fly ash were Ca(OH)_2_, CaCO_3_, SiO_2_, and Al_2_O_3_, which was consistent with the XRF of MSWI fly ash. Compared with MSWI fly ash, the diffraction peaks of Ca(OH)_2_ in the AAS solidified body (25% GGBS dosage) disappeared, and the diffraction peaks of CaCO_3_ and SiO_2_ were weakened, but the diffraction peaks of C-A-S-H and N-A-S-H formed. C-A-S-H and N-A-S-H gels were the main products of AAS, indicating that GGBS and a small amount of MSWI fly ash were involved in the polymerization reaction.

The physical phase composition of the SEAAS solidified body (1% Na_2_S dosage + 25% GGBS dosage) was similar to that of the AAS solidified body, but the diffraction peaks of C-A-S-H and N-A-S-H were slightly enhanced in the SEAAS solidified body, indicating that the S^2−^-enriched modification slightly enhanced the degree of polymerization reaction in the SEAAS solidified body. In addition, the SEAAS solidified body had diffraction peaks of PbS and CdS, and the CdS diffraction peak was more distinctive than the PbS diffraction peak, indicating that a large amount of Cd and a small amount of Pb not solidified by AAS were solidified by SEAAS in the form of sulfide.

#### 3.6.3. FT-IR Analysis

(1)MSWI fly ash

Figure 9 shows the FT-IR spectra of the original MSWI fly ash sample and the solidified bodies. According to the findings, there were apparent characteristic absorption peaks at 3415, 1618, 1421, 1156, 873, and 676 cm^−1^ in the FT-IR spectrum of the MSWI fly ash sample, corresponding to the stretching vibration peak of OH^−^, the bending vibration peak of OH^−^, the stretching vibration peak of O-C-O bond, the asymmetric stretching vibration peak of Si-O-Si/Al, the symmetric stretching vibration peak of Al-OH, and the bending vibration peak of Al-O-Si, respectively. These results indicate that the main components of the MSWI fly ash were silicate, aluminate, and carbonate, which were consistent with the XRF and XRD of MSWI fly ash.

(2)AAS solidified body

Compared with the MSWI fly ash sample, the AAS solidified body (25% GGBS dosage) also showed absorption peaks corresponding to the characteristic structure of MSWI fly ash at 3415, 1618, 1425, 874, and 655 cm^−1^. The difference was that the asymmetric stretching vibration peak of Si-O-Si/Al of the AAS solidified body shifted from 1156 to 1013 cm^−1^, showing a broadening trend. This indicates that AAS affected the molecular structure of MSWI fly ash. The polymerization products of AAS may promote the chemical bonding of heavy metals in the fly ash and result in the formation of a complex cationic layer around Al-O, thus leading to the widening of the absorption peak of Si-O-Si/Al. Moreover, the shift in vibration frequency of the absorption peak also indicates that the chemical combination of Pb and Cd in MSWI fly ash with the AAS significantly influenced the vibration frequency of the chemical bonds.

(3)SEAAS solidified body

The infrared absorption peaks of the SEAAS solidified body (1% Na_2_S dosage + 25% GGBS dosage) were similar to those of the AAS solidified body, except that a new absorption peak was generated at 797 cm^−1^ due to the vibration of sulfide precipitates. This indicates that SEAAS had not only the ability of traditional AAS to bond heavy metal ions chemically, but also to trap heavy metal ions to form stable sulfide precipitates. Sulfide precipitation and chemical bonding are two kinds of heavy-metal solidification characteristics that do not interfere with each other, so SEAAS has better solidification performance for Pb and Cd.

## 4. Solidification Mechanism Analysis

### 4.1. Solidification Mechanism of the SEAA for Pb and Cd in MSWI Fly Ash

(1)Effects of S^2−^ affinity on the solidification performance of the SEAA

The affinities of the cations of some elements to S^2−^ followed the sequence of Cd > Hg > Ag > Ca > Cu > Sb > Sn > Pb > Zn > Ni > Co > Fe > As [35]. This affinity sequence represents the reaction order of S^2−^ with different heavy metal ions in the solution. According to this rule, Cd had the strongest affinity to S^2−^, while Pb had a relatively weak affinity. The fact that S^2−^ in SEAA could preferentially capture Cd in MSWI fly ash and react with Cd^2+^ to generate CdS precipitate, which had a higher priority than PbS precipitation generation, explaining SEAA had a greater solidification performance for Cd than for Pb.

(2)Effects of S^2−^-enriched modification on the solidification performance of the SEAA

The distribution of each component(H_2_S, HS^−^, and S^2−^) in the solution of Na_2_S as a function of the pH value is as follows: when pH < 7.0, H_2_S was dominant in the solution, whereas for pH > 7.0, HS^−^ was dominant in the solution; when pH > 13.5, S^2−^ was the dominant component, and there was no generation of volatile H_2_S gas [36]. The pH value of SEAA was pH > 13.5, providing a highly alkaline environment for the precipitation of S^2−^ in Na_2_S, thus completely avoiding the loss of a large amount of S^2−^ due to the harmful volatile H_2_S gas generated by its dissolution. This not only satisfied the environmental benefits, but also made the solution rich in S^2−^. Moreover, there were some silica-aluminum oxide components in MSWI fly ash. These components could be activated by SEAA to a certain extent, thus generating a small amount of [AlO_4_] tetrahedral network structure.

To sum up, compared with traditional Na_2_S solidifying agents, SEAA had a better solidification performance for Pb and Cd heavy metals in MSWI fly ash for the following reasons: (1) SEAA provided a highly alkaline environment, enabling the dissolution of a large amount of S^2−^ in Na_2_S; hence, S^2−^ better captured Pb and Cd ions to form sulfide precipitates. (2) When MSWI fly ash was activated by SEAA, a small amount of electronegative [AlO_4_] tetrahedral network structure could be generated, which could bond heavy metal ions chemically.

### 4.2. Solidification Mechanism of the SEAAS for Pb and Cd in MSWI Fly Ash

(1)Effects of ionic radius on the solidification performance of AAS of Pb and Cd

The order of the ionic radii of K, Na, Pb, and Cd was K^+^ (0.138 nm) > Pb^2+^ (0.119 nm) > Na^+^ (0.102 nm) > Cd^2+^ (0.097 nm). The ionic radius affected the ability of ionic bond coordination to some extent. Therefore, the order of their ionic radii could explain the better solidification performance of AAS for Pb than for Cd in MSWI fly ash. Al^3+^ showed electronegativity after combining with four O^2−^ in the structure of the AAS polymerization product. To balance the electrovalence, some cations, such as Na^+^ and K^+^, participate in the formation of structural monomers and play a role in balancing the electrovalence [37]. The radius of Pb^2+^ was between the radii of Na^+^ and K^+^, which easily bond to the [AlO_4_] structure of AAS [38]. It can be considered that Pb^2+^ is more likely than Cd^2+^ to participate in the formation of polymerization products of AAS, thus replacing the sites of Na^+^ or K^+^ in the [AlO_4_] tetrahedral structure of AAS to a large extent. Finally, Pb^2+^ was effectively bonded into the network structure system of AAS in the form of an equilibrium charge. At the same time, the greater the GGBS dosage, the more [AlO_4_] structure generated by the AAS polymerization reaction, making it more effective at solidifying Pb and Cd.

(2)Effects of synergetic effect on the solidification performance of SEAAS for Pb and Cd

Figure 10 shows the solidification mechanism of Cd^2+^ and Pb^2+^ in MSWI fly ash by SEAAS. The process was divided into two stages. In the first stage, when SEAA and MSWI fly ash were mixed and stirred, most of the free Cd^2+^ in MSWI fly ash were solidified into stable CdS by S^2−^ in SEAA; however, a large number of free Pb^2+^ in MSWI fly ash were not solidified due to their weak affinity to S^2−^. In the second stage, with the dosage of GGBS into the SEAA solidifying slurry, a large number of electronegative [AlO_4_] tetrahedral network structures were generated through the polymerization reaction, making the Pb^2+^ ions that were unsolidified by S^2−^ in SEAA bond to the network structure system of SEAAS in the form of equilibrium charge. In the meantime, the sulfide precipitates formed in the first solidification stage were also effectively sealed in SEAAS gel through a physical barrier. Hence, under the synergistic effect of sulfide precipitation and chemical bonding, SEAAS could efficiently solidify Pb^2+^ and Cd^2+^ in MSWI fly ash, significantly reducing the leaching concentrations of Pb^2+^ and Cd^2+^ in solidified MSWI fly ash.

## 5. Conclusions

After the inclusion of further Na_2_S, the SEAA’s solidification performance for Pb and Cd in MSWI fly ash increased, initially noticeably and then slightly. The critical Na_2_S dosage at which the solidification performance increased significantly was a Na_2_S dosage of 1%. The solidification performance of the SEAA for Pb in MSWI fly ash was considerably weaker than that for Cd; thus, it is challenging to solve the exceeding standard problem of Pb in the low Na_2_S dosage range of 0.5~1%.The performance of SEAAS for Pb and Cd in MSWI fly ash solidification increased as the GGBS dosage was elevated, initially noticeably and then slightly. The critical GGBS dosage at which the solidification performance increased significantly was a GGBS dosage of 25%. In the low GGBS dosage range (0~25%), SEAAS could effectively solidify Pb and Cd in MSWI fly ash, and the solidification performance for Cd was better than that for Pb, which successfully compensated for the shortcomings of the poor solidification performance of traditional AAS for Cd in MSWI fly ash.The chemical and molecular composition of the SEAAS solidified body were similar to those of the AAS solidified body, except that new characteristic peaks of sulfides were detected on the XRD and FT-IR spectrum of the SEAAS solidified body. The highly alkaline environment provided by SEAA promoted the dissolution of a large amount of S^2−^ in the solvent, endowing the SEAAS with a strong ability to capture Pb and Cd to form sulfide precipitates. SEAAS not only had the capability of chemically bonding Pb and Cd using the [AlO_4_] network structure, but also had the function of capturing Pb and Cd to form stable sulfide precipitates.

## Figures and Tables

**Figure 1 materials-16-03728-f001:**
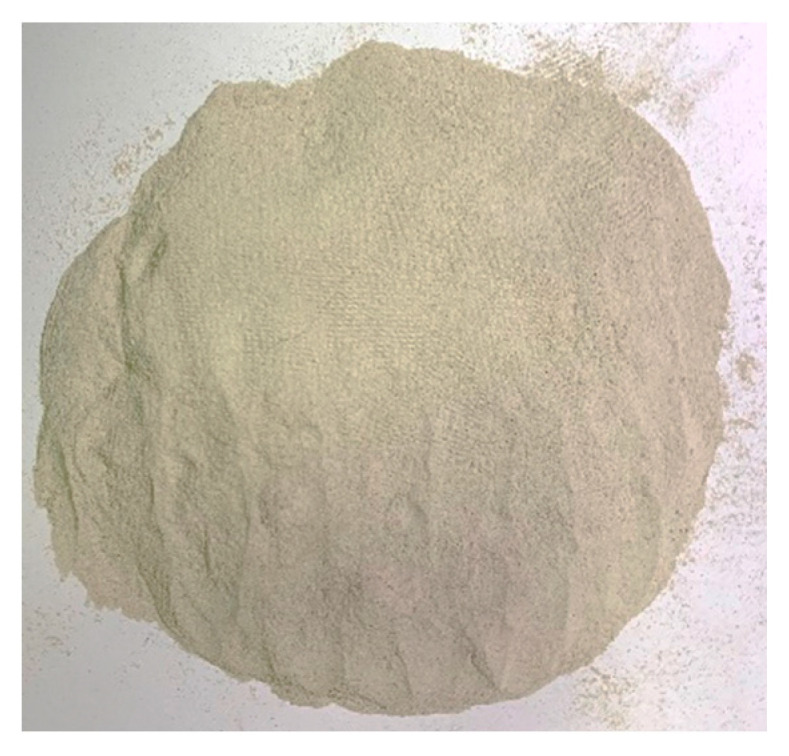
The appearance of MSWI fly ash.

**Figure 2 materials-16-03728-f002:**
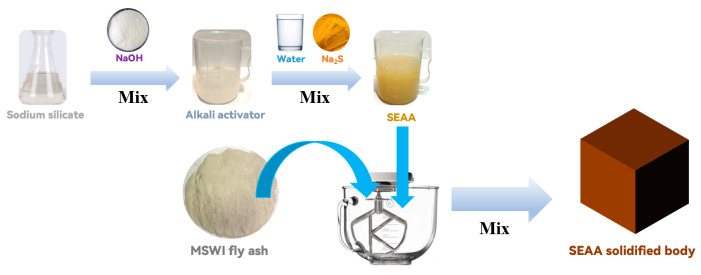
The preparation process of the SEAA solidified body.

**Figure 3 materials-16-03728-f003:**
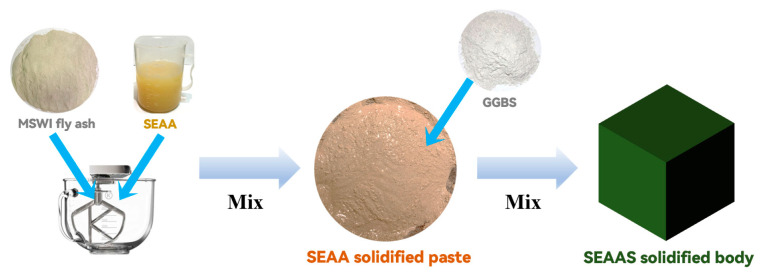
The preparation process of SEAAS solidified body.

**Figure 4 materials-16-03728-f004:**
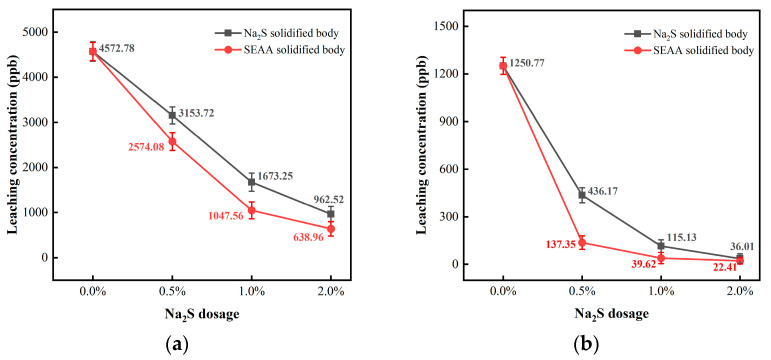
Effects of Na_2_S dosage on the solidification performance of SEAA for Pb and Cd. (**a**) Leaching concentration of Pb. (**b**) Leaching concentration of Cd.

**Figure 5 materials-16-03728-f005:**
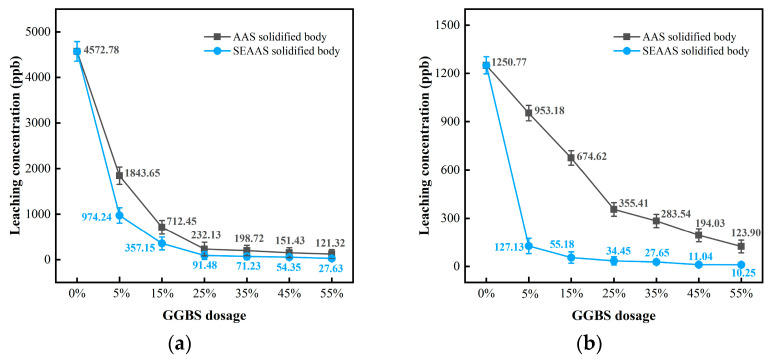
Effects of GGBS dosage on the solidification performance of SEAAS for Pb and Cd. (**a**) Leaching concentration of Pb. (**b**) Leaching concentration of Cd.

**Figure 6 materials-16-03728-f006:**
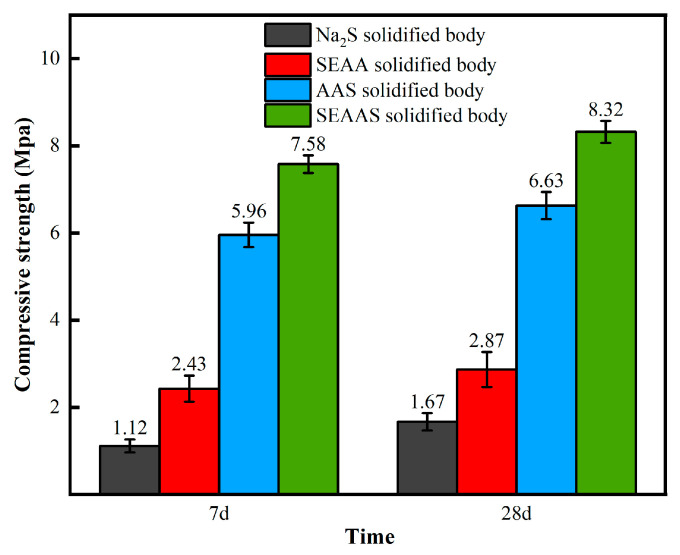
The compressive strength of different fly ash solidified bodies.

**Figure 7 materials-16-03728-f007:**
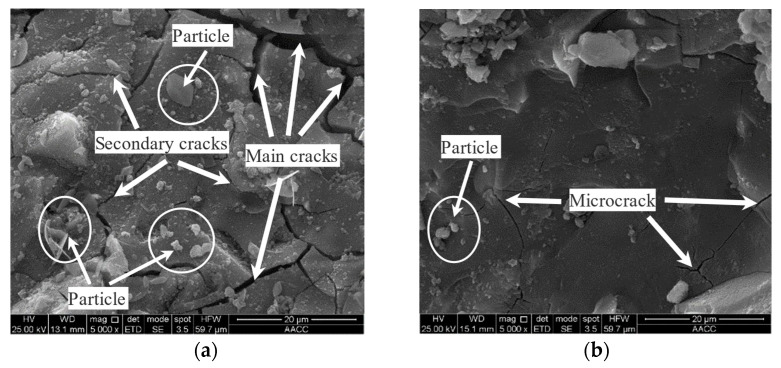
The micro-morphology of the solidified bodies. (**a**) AAS solidified body (25% GGBS dosage). (**b**) SEAAS solidified body (1% Na_2_S dosage + 25% GGBS dosage).

**Figure 8 materials-16-03728-f008:**
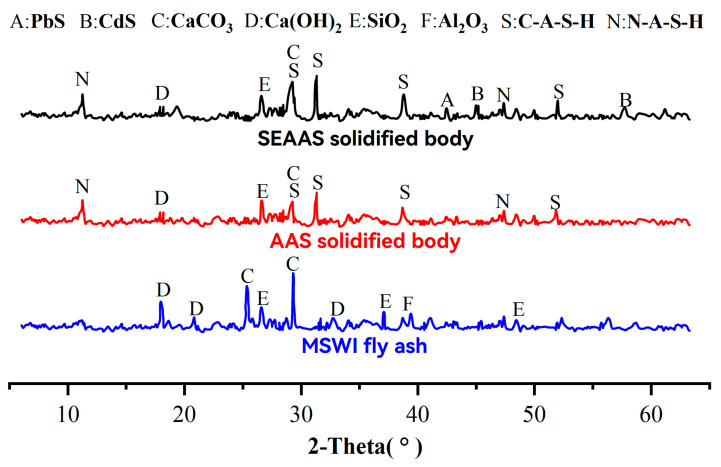
The XRD of MSWI fly ash and solidified bodies.

**Figure 9 materials-16-03728-f009:**
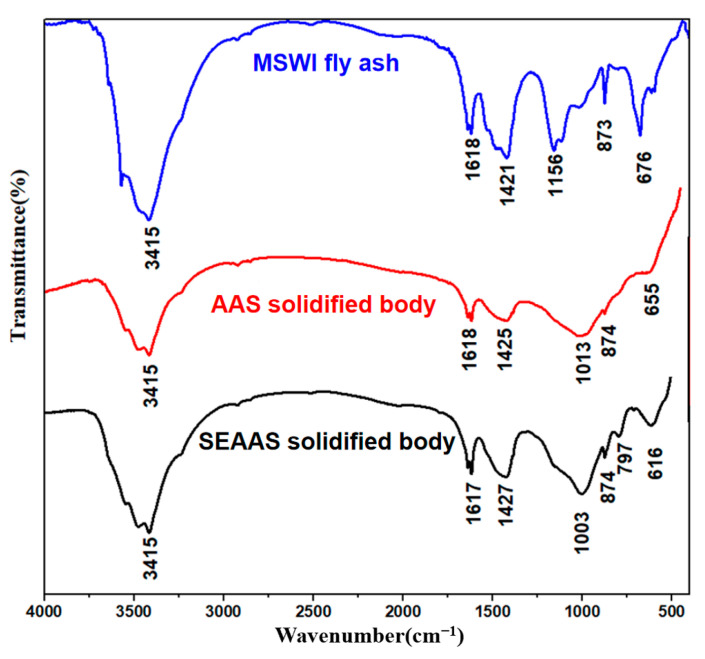
FT-IR spectra of the MSWI fly ash and solidified bodies.

**Figure 10 materials-16-03728-f010:**
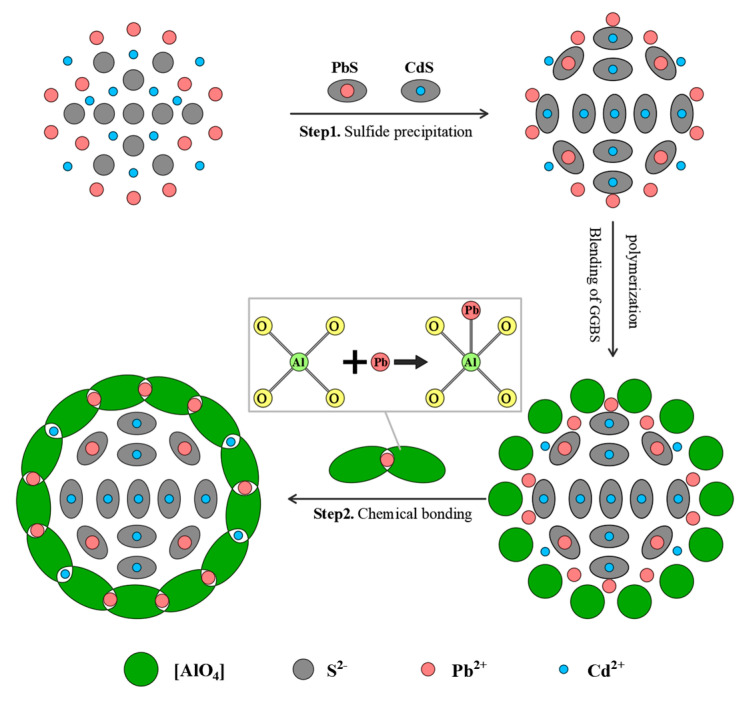
Solidification mechanism of Cd^2+^ and Pb^2+^ in MSWI fly ash by SEAAS.

**Table 1 materials-16-03728-t001:** Chemical composition of materials/(wt%).

	CaO	SiO_2_	A1_2_O_3_	Fe_2_O_3_	MgO	Na_2_O
MSWI fly ash	38.62	17.21	7.02	1.79	4.01	3.11
GGBS	34.65	31.35	18.65	0.57	9.31	0.41

**Table 2 materials-16-03728-t002:** Leaching toxicity of heavy metal ions in MSWI fly ash/(ppb).

	Be	Cr	Ni	Cu	Zn	Se	Cd	Ba	Pb
MSWI fly ash	0.65	241.92	26.09	3080.07	15,652.9	14.42	1250.77	2602.24	4572.78
GB 5085.3-2007	20	4500	500	40,000	100,000	100	150	25,000	250

**Table 3 materials-16-03728-t003:** The specific parameters of the chemical reagents.

Chemical Reagent	Purity Specification	Manufacturer
Sodium sulfide	Analytical Reagent	Fu Chen (Tianjin) Chemical Reagent Co., Ltd., Tianjin, China
Glacial acetic acid	Analytical Reagent	Fu Chen (Tianjin) Chemical Reagent Co., Ltd., Tianjin, China
Sodium hydroxide	Analytical Reagent	Xilong Chemical Co., Ltd., Shantou, China

**Table 4 materials-16-03728-t004:** The pH values of the alkali activator and SEAA.

Activator	PH Value
Alkali activator	13.81
SEAA	13.94

**Table 5 materials-16-03728-t005:** XRF of MSWI fly ash and solidified bodies.

Chemical Composition (wt%)	MSWI Fly Ash	AAS Solidified Body	SEAAS Solidified Body
CaO	38.62	43.23	42.69
SiO_2_	17.21	23.18	23.26
CO_3_	21.9	6.12	5.88
Al_2_O_3_	7.02	10.56	9.83
MgO	4.01	2.12	2.06
Na_2_O	3.11	8.95	10.03
Fe_2_O_3_	1.79	1.13	1.02
K_2_O	1.28	0.95	0.79
S	0.56	0.73	3.24
PbO	0.0847	0.0371	0.0306
CdO	0.0432	0.0315	0.0237
Pb	0.0155	0.0013	0.0097
Cd	0.0102	0.0087	0.0205

## Data Availability

Not applicable.

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
