# Peer review of "Solidification Mechanism of Pb and Cd in S2−-Enriched Alkali-Activated Municipal Solid Waste Incineration Fly Ash"

_materials, 2023, doi:10.3390/ma16103728_

Round 1

Reviewer 1 Report

The paper has a clear contribution to the field. Few minor comments should be addressed by the authors:

- In line number 71-72, which reference do the authors refer to when they say "by the authors", it is not clear.

-In line number 46-47, reference should be added.

-In line number 117-118, photographs of MSWI fly ash samples in this study should be shown for better demonstration.

-In line number 188 -200, a diagram that demonstrate the process should be added for better clarification.

- Few typos should be avoided, for example:

- Delete the doupled %% in line number 246.

-Unify the symbols, for example, line 278 and line 283, the number 2 is small in one and large in the other.

- In line 580 (reference 27) delete the word "In" because it is not part of the paper's title. Other references should be revised to avoid this kind of mistakes.

Author Response

Response to Reviewer 1 Comments

Reviewer 1:

The paper has a clear contribution to the field. Few minor comments should be addressed by the authors:

Remark 1: In line number 71-72, which reference do the authors refer to when they say "by the authors", it is not clear.

Response: Thank you for your careful question.

We have given you a misunderstanding due to our unclear expression. The ' author ' refers to the research carried out by our team. In the previous exploratory experiment, our team found that the curing performance of AAS on MSWI fly ash Cd had shortcomings, which was reflected in chapter 3.2.

Remark 2: In line number 46-47, reference should be added.

Response: Thank you for your careful advice. We have added references here.

Solidification by cementitious materials is considered the most suitable method for MSWI fly ash because of its simple treatment and sound economic and environmental effects[14].

Remark 3: In line number 117-118, photographs of MSWI fly ash samples in this study should be shown for better demonstration.

Response: Thank you for your careful advice. We have added a photograph of the MSWI fly ash sample here.

Figure 1. The appearance of MSWI fly ash

Remark 4: In line number 188 -200, a diagram that demonstrate the process should be added for better clarification.

Response: Thank you for your careful advice. We have added two diagrams that demonstrate the preparation process of solidified body.

Figure 2. The preparation process of SEAA solidified body

Figure 3. The preparation process of SEAAS solidified body

Remark 5: Few typos should be avoided, for example:

Delete the doupled %% in line number 246.

Unify the symbols, for example, line 278 and line 283, the number 2 is small in one and large in the other.

Response: Thank you for your careful advice. We have corrected the two typos.

Remark 6: In line 580 (reference 27) delete the word "In" because it is not part of the paper's title. Other references should be revised to avoid this kind of mistakes.

Response: Thank you for your careful advice. We have deleted the word "In" here. We have also checked other references to ensure accuracy.

Reviewer 2 Report

in the attachment

Reviewer 3 Report

Solidification mechanism of Pb and Cd in S2--enriched alkali-2 activated municipal solid waste incineration fly ash

by Qi Xue, Yongsheng Ji, Zhanguo Ma, Zhongzhe Zhang and Zhishan Xu

in Materials MDPI

General comments

·                  In overall, I found the work very interesting and in line with the Journal scope. Presentation and writing style is good with some lapses in style. Structure is clear.

·                  Title: appropriate

·                  Abstract: it well written however the importance of the study should be underlined and the main implications of the study findings.

·                  Introduction: Regarding the use of Na2S are there any previous studies?

·                  Materials and methods: Good description, coherent and detailed. However possible the “influences” may be corrected with “effects”?

·                  References:>73% οf References are up-to-date.

Specific comments (list not exhaustive)

Introduction: If it possible, to add some data on the amount of MSWI fly ash globally?

Materials: GGBS and MSWI fly ash where do they come from?

L156: what is the concentration of acetic acid used in this study?

Perhaps a schematic diagram(s) could better depict the (mixing) process for the preparation of the final(s) solidified body(ies) and also present what are the solidification bodies tested.

Results: Why was 2% of Na2S the maximum dosage?

Figure 1(a), (b) (the same for Fig.2) maybe if you consider to underline the limitations (standard level) regarding Pb and Cd? Please use the same y-axis for all figures.

It is not clear the purpose of Na2S solidified body measurements. Please clarify.

L236 (and within the manuscript) maybe it is better to use effect instead of influences?

Fig. 3 do you perform one measurement or multiple ones?

L360: 8ΜPa is considered a low compressive strength, how mechanical properties improved?

Fig. 4 please add the GGBS and Na2S dosage

L382: Could you calculate the concentration of OH- in the solution during polymerization?

Do you consider to perform also an XRD analysis (to prove probably sulfide precipitation)?

L420/ Fig. 5 please include all bodies (AAS and SEAAS)

L424: Please provide Reference(s) for solubility products

L470: Could findings be linked with leaching measurements?

Figure 7: Isn’t it is a suggested solidification process?

FTIR results showed sulfide precipitation (through vibration at 797cm-1) However L481: “..Pb2+ ions that were unsolidified by S2- in the SEAA be bonded to the network structure system of the SEAAS in the form of equilibrium charge.” Could you prove this claim?

 On the basis of the above, my recommendation is a Minor revision and I hope the authors define my perspective useful in improving their manuscript.

Reviewer 4 Report

In paragraph 2: Materials and Methods, parameters without scientific basis were established. Perhaps a correct way would be to present an experimental design to evaluate the alkali contents.

In paragraph 3, figure 4 a) and b) should be better discussed with regard to densification and cracking

Author Response

Reviewer 4:

Remark 1: In paragraph 2: Materials and Methods, parameters without scientific basis were established. Perhaps a correct way would be to present an experimental design to evaluate the alkali contents.

Response: Thank you for your careful advice.

We have added the pH test of the activator (characterization of OH- concentration) and the XRF analysis of the solidified body (characterization of Na2O content) in 2.5 of the paragraph 2. The results of these tests were discussed to evaluate the alkalinity of the activator and the alkali content of the solidified body. The additions were as follows:

2.5.3 Effects of S2--enriched modification on the alkalinity of the activator

(1) Preparation of testing fluid

The SEAA with the best Na2S content was selected, and the alkali activator and SEAA were prepared according to 2.2 to study the effects of S2--enriched modification on the alkalinity of the activator.

(2) Determination of pH

The pH value of the activator was determined by an acidometer (Ritz, PHB-5, Shanghai). The pH range of the acidometer was -2~18, and the accuracy was 0.01.

2.5.6 Effects of SEAAS on the Chemical and molecular composition of MSWI fly ash

(1) Mixing ratio and preparation of samples

The SEEAS solidified body with the same ratio as 2.5.4 was prepared. The original MSWI fly ash and the AAS solidified body with the same GGBS dosage were used as a reference to study the effects of SEAAS on the chemical and molecular composition of MSWI fly ash.

The original MSWI fly ash and the solidified body with curing ages of seven days were dried at 65℃ for 24 h. They were then ground and screened through a 325-mesh sieve. The screened powder was pressed into flat test pieces of 1~10 μm thickness by the KBr pellet pressing method as the test samples.

(2) XRF analysis

An X-ray fluorescence spectrometer (Bruker, S8 TIGER, Germany) was used for the analysis of the chemical composition of the test samples.

3.3 Effects of S2--enriched modification on the alkalinity of the activator

Table 4 shows the pH values of the alkali activator and SEAA. As can be seen, the pH value of the alkali activator was 13.81, and the pH value of SEAA was 13.94. Compared with the alkali activator, the concentration of OH- in SEAA increased by 0.225 mol / L.

Table 4. The pH values of the alkali activator and SEAA

Activator

PH value

Alkali activator

13.81

SEAA

13.94

The reason for the higher OH- concentration of SEAA was that Na2S dissolved in water could form sodium hydroxide, as shown in Eq. (2). Therefore, compared with the AAS activator with the same sodium silicate modulus, the concentration of OH- in SEAA was increased, which provided a higher alkaline environment for the dissolution of S2- and the polymerization reaction of powders (GGBS and MSWI fly ash).

Na2S + H2O = NaHS + NaOH                              (2)

3.6.1 XRF analysis

Table 4 shows the XRF of MSWI fly ash and solidified bodies. The main components of MSWI fly ash were CaO, SiO2, Al2O3, etc. Pb and Cd were present in MSWI fly ash in the form of most oxides and a small portion of non-oxides. Compared with the MSWI fly ash, the content of CaO, SiO2, Al2O3, and Na2O in the AAS solidified body (25% GGBS dosage) increased, and the content of Pb, Cd, PbO, and CdO decreased.

Compared with AAS solidified sample, the contents of CaO, SiO2, and Al2O3 in the SEAAS solidified body (1% Na2S dosage + 25% GGBS dosage) were approximately the same. The difference was that its Na2O content has increased, and the content of sulfur has increased significantly, which indicates that S2--enriched modification increases the alkali content and sulfur content in the solidified body. In addition, the increase of sulfur content in the SEAAS solidified sample was accompanied by an increase in the content of Cd and Pb due to the formation of sulfide precipitates. The increase of Cd was greater, indicating that SEAAS has a stronger ability to capture Cd to form stable sulfides.

Table 4 The XRF of MSWI fly ash and solidified bodies

Chemical

Composition

(wt%)

MSWI fly ash

AAS solidified body

SEAAS solidified body

CaO

38.62

43.23

42.69

SiO2

17.21

23.18

23.26

CO3

21.9

6.12

5.88

Al2O3

7.02

10.56

9.83

MgO

4.01

2.12

2.06

Na2O

3.11

8.95

10.03

Fe2O3

1.79

1.13

1.02

K2O

1.28

0.95

0.79

S

0.56

0.73

3.24

PbO

0.0847

0.0371

0.0306

CdO

0.0432

0.0315

0.0237

Pb

0.0155

0.0013

0.0097

Cd

0.0102

0.0087

0.0205

Remark 2: In paragraph 3, figure 4 a) and b) should be better discussed with regard to densification and cracking

Response: Thank you for your careful advice.

We have labeled the cracks and particles in these two figures, and discussed the size of the cracks and the compactness of the gel in more detail. The specific changes are as follows:

3.5 Effects of SEAAS on the micro-morphology of MSWI fly ash

Figure 7 presents the micro-morphology of the solidified bodies. Figure 7(a) illustrates the micro-morphology of the AAS solidified body (25% GGBS dosage), and Figure 7(b) shows the micro-morphology of the SEAAS solidified body (1% Na2S dosage + 25% GGBS dosage).

(1) Micro-morphology of AAS solidified body

It can be seen from Figure 7(a) that some irregular particles were scattered on the surface of the AAS solidified body, which indicated that a small amount of MSWI fly ash and GGBS particles in the AAS solidified body did not fully participate in the polymerization reaction. At the same time, there were also a large number of main cracks (1~2μm) and secondary cracks (0.5~1μm) in the micro-morphology of the sample. The main cracks were crisscrossed and irregularly extend everywhere, affecting the solidified body's integrity. The end of the secondary cracks continued to expand and extend, and its extension direction was in the same direction as the main cracks, which had not only a negative effect on the mechanical properties of the solidified body but also weakened the physical barrier effect of AAS on some free heavy metals in MSWI fly ash.

(2) Micro-morphology of SEAAS solidified body

As can be seen from Figure 7(b), compared with the AAS solidified body, most of the particles on the surface of the SEAAS solidified body disappeared, and the sample showed a uniform and dense gel. At the same time, there were no main cracks and secondary cracks in the micro-morphology of the sample, only a small amount of microcrack (< 0.5μm), and its extension and penetration were significantly reduced. Combined with the results of 3.3, the SEAA had a higher OH- concentration, which made the polymerization reaction between GGBS and SEAA more complete, thus producing a more compact gel structure. Therefore, the SEAAS solidified body had better mechanical properties and a more prominent physical barrier effect on free heavy metals.

Round 2

Reviewer 2 Report

Thank you for your response. I appreciate your work but  I have still some comments:

- to put more attention into statistic analysis (more details according to figure description),

- to improve part "the definition of terms" (visual and language side) eg. it is better to write the mass ratio of water/powder than "The ratio of water weight /powder weight". You didn't add all terms (MSWI, GGBS etc.....)

- to improve part "Materials and Method". It should be written more clearly and with more details.

- to improve stylistic site eg. There is "According to HJ/T 300-2007 the leaching concentrations of heavy metal ions in  the MSWI fly ash leaching solution were analyzed by an ICP-MS laser denudation-plasma  mass spectrometer, and Table 2 shows the results."

but it is more clearly and shortly:

"According to HJ/T 300-2007 the leaching concentrations of heavy metal ions in  the MSWI fly ash leaching solution were analyzed by an ICP-MS laser denudation-plasma  mass spectrometer (Table 2) "

- There is still lack of consistent names: Sodium silicate and  NaOH -> sodium silicate and sodium hydroxide

Author Response

Response to Reviewer

On behalf of my co-authors, we thank you very much for giving us an opportunity to revise our manuscript, and we also appreciate reviewers very much for their positive and constructive comments and suggestions on our manuscript entitled “Solidification mechanism of Pb and Cd in S2--enriched alkali-activated municipal solid waste incineration fly ash”(Manuscript ID: materials-2319304).

We revised the manuscript according to these comments and suggestions. In general, we have tried our best to revise our manuscript and provide the point-by-point responses. All changes were highlighted the modified or supplemented parts in blue. We have sent the revised manuscript, and a version containing all the changes to be visible. Point by point responses to the reviewers’ comments are listed below this letter.

-------------------------------------------------------------------------------------------------------

Response to Reviewer 2 Comments

Reviewer 2:

Thank you for your response. I appreciate your work but I have still some comments:

Remark 1: to put more attention into statistic analysis (more details according to figure description)

Response: Thank you for your careful advice. We have provided more details in the figure description to ensure better clarity and understanding of the data. Thank you again for your valuable input.

3.1. Effects of Na2S dosage on the solidification performance of SEAA for Pb and Cd

(1) Solidification performance for Pb

Figure 4(a) shows the effects of Na2S dosage on the solidification performance of SEAA for Pb. According to the findings, the original leaching concentration of Pb in MSWI fly ash was noted as 4572.78 ppb. When the Na2S dosages were 0.5%, 1%, and 2%, the leaching concentrations of the Na2S solidified body (control group) were determined as 3153.72, 1673.25, and 962.52 ppb, respectively, and the corresponding solidification rates were 31.03%, 63.44%, and 78.95% respectively.

When the Na2S dosage was 0.5%, 1%, and 2%, the SEAA solidified body leaching concentrations were 2574.08, 1047.56, and 638.96 ppb, respectively, and the corresponding solidification rates were 43.71%, 77.09%, and 86.02% respectively. The solidification rates of the SEAA solidified body increased by 12.68%, 13.68%, and 7.07% compared with those of the Na2S solidified body at each dosage level. These results showed that under the same Na2S dosage, the leaching concentration of Pb in the SEAA solidified body was slightly lower than that in the control group, indicating that the solidification performance of SEAA for Pb in MSWI fly ash was slightly better than Na2S solidified body.

There was a negative correlation between Na2S dosage and Pb leaching concentration of SEAA solidified body, especially in the range of 0% to 1% Na2S dosage, indicating that the increase of Na2S dosage could reduce the Pb leaching concentration of SEAA solidified body. However, when the Na2S dosage was 2%, the leaching concentration of Pb in the SEAA solidified body was still higher than the standard limit of 250 ppb required by GB 5085.3-2007.

(2) Solidification performance for Cd

Figure 4(b) shows the effects of Na2S dosage on the solidification performance of SEAA for Cd. The original leaching concentration of Cd in MSWI fly ash was found to be 1250.77 ppb. When the Na2S dosages were 0.5%, 1%, and 2%, the leaching concentrations of the Na2S solidified body (control group) were 436.17, 115.13, and 36.01 ppb, respectively, and the corresponding solidification rates were 65.13%, 90.80%, and 97.12% respectively.

When the Na2S dosages were 0.5%, 1%, and 2%, the leaching concentrations of SEAA solidified body was 137.35, 39.62, and 22.41 ppb, respectively, and the corresponding solidification rates were 89.02%, 96.83%, and 98.21% respectively. The solidification rates of the SEAA solidified body increased by 23.89%, 6.04%, and 1.09% compared with those of the Na2S solidified body at each Na2S dosage level. These results indicated that the solidification performance of SEAA for Cd in MSWI fly ash was much better than control group.

There was a stronger negative correlation between Na2S dosage and Cd leaching concentration of SEAA solidified body, especially in the range of 0% to 0.5% Na2S dosage, indicating that the increase of Na2S dosage could significantly reduce the Cd leaching concentration of SEAA solidified body. When the Na2S dosage was 0.5%, the leaching concentration of Cd in the SEAA solidified body was comparable to that in the research group with a Na2S dosage of 1%, and was lower than the standard limit of 150 ppb required by GB 5085.3-2007.

3.2 Effects of GGBS dosage on the solidification performance of SEAAS for Pb and Cd

(1) Solidification performance for Pb

Figure 5(a) presents the effects of GGBS dosage on the solidification performance of the SEAAS for Pb. As can be seen, when the GGBS dosage was 5%, 15%, and 25%, the leaching concentrations of Pb from the AAS solidified body (control group) were 1843.65, 712.45, and 232.13 ppb, respectively, and the corresponding solidification rates were 59.68% %, 84.42%, and 94.92% respectively. With the further increase of GGBS dosage to 35%, 45%, and 55%, the leaching concentration of Pb from the AAS solidified body decreased further but tended to be gentle. When the GGBS dosage was 55%, the Pb leaching concentration of the AAS solidified body was 21.32 ppb, and the solidification rate was only 3.12% higher than that of the AAS solidified body with a GGBS dosage of 25%.

When the GGBS dosages were 5%, 15%, and 25%, the leaching concentrations of Pb from the SEAAS solidified body was 974.24, 357.15, and 91.48 ppb, respectively. The corresponding solidification rates were 80.88%, 90.01%, and 97.99%, respectively, which increased by 30.78%, 20.15%, and 13.06% compared with those of AAS solidified body at each dosage level. These results indicated that SEAAS was slightly better than AAS for Pb solidification of MSWI fly ash.

When the GGBS dosage was in the range of 0%~25%, there was a strong negative correlation between GGBS dosage and the Pb leaching concentration of SEAAS solidified body, indicating that the increase of GGBS dosage could significantly reduce the Pb leaching concentration of SEAA solidified body, so that the leaching concentration of SEAAS solidified body with 25 % GGBS dosage was lower than the standard limit of 250 ppb required by GB 5085.3-2007.

With the further increase of GGBS dosage to 35%, 45% and 55%, there was a weak negative correlation between GGBS dosage and Pb leaching concentration of SEAAS solidified body, indicating that the Pb leaching concentration of SEAAS solidified body was not significantly reduced in these GGBS dosage ranges.

(2) Solidification performance for Cd

Figure 5(b) shows the effects of GGBS dosage on the Cd solidification performance of the SEAAS. As can be seen, when the GGBS dosages were 5%, 15%, and 25%, the leaching concentration of the AAS solidified body (control group) showed an obvious decreasing trend; however, the leaching concentration of the AAS solidified body with a GGBS dosage of 25% was 355.41 ppb, which was still higher than the standard limit of 150 ppb required by specifications. With the further increase of GGBS dosage to 35%, 45%, and 55%, the leaching concentration of AAS solidified body was further reduced but tended to be moderate. When the GGBS dosage was 55%, the Cd leaching concentration of the AAS solidified body was 123.9 ppb, lower than the standard limit of 150 ppb required by specifications. Hence, AAS had poor solidification performance for Cd in MSWI fly ash.

When the GGBS dosage was in the range of 0%~15%, there was a strong negative correlation between GGBS dosage and the Cd leaching concentration of SEAAS solidified body. The Cd leaching concentration of SEAAS solidified body with 5 % GGBS dosage was 127.13 ppb, and the corresponding solidification rate was 90.00%. This leaching concentration was equivalent to the AAS solidified body with GGBS dosage of 55% (123.9 ppb), lower than the standard limit of 150 ppb required by GB 5085.3-2007,revealing that the SEAAS had excellent solidification performance for Cd in MSWI fly ash.

With the further increase of GGBS dosage to 25%, 35%, 45% and 55%, there was a weak negative correlation between GGBS dosage and Cd leaching concentration of SEAAS solidified body, indicating that the Cd leaching concentration of SEAAS solidified body was not significantly reduced in these GGBS dosage ranges.

Remark 2: to improve part "the definition of terms" (visual and language side) eg. it is better to write the mass ratio of water/powder than "The ratio of water weight /powder weight". You didn't add all terms (MSWI, GGBS etc.....)

Response: Thank you for your comments on our manuscript. We appreciate your feedback regarding the improvement of the “definition of terms” section in our paper. We agree that using ‘the mass ratio of water to powder’ instead of “the ratio of water weight / powder weight” would be more clear and concise. We have revised the section accordingly. Additionally, we have ensured that all relevant terms, including MSWI and GGBS, are included in the definitions for clarity and precision. Thank you for bringing these issues to our attention.

The definition of terms

MSWI fly ash  Municipal solid waste incineration fly ash

GGBS  Ground granulated blast-furnace slag

Alkalia activator  Water solution sodium silicate+NaOH

SEAA  Alkalia activator + Na2S

SEAA solidified body  MSWI fly ash + SEAA  

SEAAS solidified body  MSWI fly ash + SEAA+ GGBS

AAS solidified body  MSWI fly ash + alkalia activator + GGBS

Na2S solidified body  MSWI fly ash + Na2S

AAS   Alkalia activator + GGBS        

Silicate modulus  The molar ratio of SiO2/Na2O

Water-binder ratio  The mass ratio of water/powder

Solidified paste  Uncondensed hardened solidified body

Remark 3: to improve part "Materials and Method". It should be written more clearly and with more details.

Response: Thank you for your comments on our manuscript. We appreciate your feedback regarding the “Materials and Methods” section of our paper. We agree that this section should be written more clearly and with more details. We have merged the Preparation of heavy metal leaching agent into Chapter 2.4 to ensure that detailed descriptions of all detection methods are included in the chapter. We have supplemented the information on the accuracy of all the microscopic detection methods used in the study. Thank you again for your valuable input.

2.4.1 Effects of Na2S dosage on the solidification performance for Pb and Cd by the SEAA

(1) Mixing proportion and preparation of specimens

The SEAA solidified bodies with different Na2S dosages at a water-binder ratio of 0.4 were prepared. In different groups, the mass of fly ash was constant, and the dosages of Na2S were 0%, 0.5%, 1%, and 2% of the mass of fly ash. The alkali-activator's solid content (mass sum of SiO2 and Na2O) was fixed at 17.5% of the fly ash mass.

The preparation process of the SEAA solidified body was shown in Figure 2. The SEAA was prepared according to 2.2, and then the SEAA solidified bodies were prepared according to subsection 2.3 by mixing the SEAA with MSWI fly ash. By taking the Na2S solidified bodies with the same water-binder ratio and the same Na2S dosage as the control group, respectively, the effects of Na2S dosage on the solidification performance for Pb and Cd by the SEAA were investigated.

Figure 2. The preparation process of SEAA solidified body

(2) Preparation of heavy metal leaching agent

Solidified body cured for seven days was dried for 24 h at 65℃, then crushed into disintegrating slag of solidified body with particle size ≤ 0.15 mm. According to the HJ/T300-2007, the disintegrating slag of solidified body (20.0g) was weighed and put into a polyethylene bottle containing 0.3mol / L glacial acetic acid diluent (400ml), and the bottle was turned over and s at 30 r/min for 18 h at 23±2℃. After standing for 6 h, 10 mL of solution was extracted. The extract was filtrated through a mesh sieve of 0.8 μm to obtain the heavy metal leaching solution of the MSWI fly ash solidified body.

(3) Determination of heavy metal contents

ICP-MS laser denudation-plasma mass spectrometer (Agilent Technologies, NWR 213-7900, America) was used to measure the Cd and Pb leaching concentrations of l heavy metal leaching agent to analyze the solidification performance of Pb and Cd in MSWI fly ash by the solidification materials.

2.5.6 Effects of SEAAS on the Chemical and molecular composition of MSWI fly ash

(1) Mixing proportion and preparation of samples

The SEEAS solidified body with the same ratio as 2.5.4 was prepared. The original MSWI fly ash and the AAS solidified body with the same GGBS dosage were used as a reference to study the effects of SEAAS on the chemical and molecular composition of MSWI fly ash.

The original MSWI fly ash and the solidified body with curing ages of seven days were dried at 65℃ for 24 h. They were then ground and sieved through a 325-mesh sieve, and the sieved powder was used as test samples for XRF and XRD analysis. The sieved powder was mixed with KBr in a ratio of 1:8 and then pressed into flat specimens of 1 to 10 μm as FT-IR test samples using a hydraulic pelletizer with a 5 Tons load capacity.

(2) XRF analysis

X-ray fluorescence spectrometer (Bruker, S8 TIGER, Germany) was used to determine the chemical composition and elemental content of the test samples.

(3) XRD analysis

The mineral composition of the test samples was analyzed using an X-ray diffractometer (Bruker, D8 Advance, Germany) with diffraction angles ranging from 5° to 65° (2θ) in steps of 0.020° at a rate of 2°/min, and Cu-Ka radiation were applied.

(4) FT-IR analysis

The vibration patterns and molecular structures of the test samples were analyzed using an infrared spectrograph (Bruker, VERTEX 80v, Germany). Each sample was scanned 32-times with a resolution of 2 cm-1, and the wavelength range was between 400 and 4000 cm-1.

Remark 4: to improve stylistic site eg. There is "According to HJ/T 300-2007 the leaching concentrations of heavy metal ions in the MSWI fly ash leaching solution were analyzed by an ICP-MS laser denudation-plasma mass spectrometer, and Table 2 shows the results."

but it is more clearly and shortly:

"According to HJ/T 300-2007 the leaching concentrations of heavy metal ions in  the MSWI fly ash leaching solution were analyzed by an ICP-MS laser denudation-plasma  mass spectrometer (Table 2) "

Response: Thank you for your careful advice. We appreciate your feedback regarding the improvement of the writing style in our paper. We agreed that the sentence you highlighted can be written more clearly and concisely. We have revised the sentence . Thank you again for your valuable input.

According to HJ/T 300-2007 [31], the leaching concentrations of heavy metal ions in  the MSWI fly ash leaching solution were analyzed by an ICP-MS laser denudation-plasma  mass spectrometer (Table 2). Cd and Pb contents in the MSWI fly ash seriously exceed the standard level by taking the concentration limits of leaching liquid pollutants in GB 5085.3-2007 [32] as the standard level, and the corresponding multiples of exceeding the standard are 8.33 times and 18.29 times respectively.

Remark 5: There is still lack of consistent names: Sodium silicate and  NaOH -> sodium silicate and sodium hydroxide.

Response: Thank you for your careful advice. We appreciate your feedback regarding the inconsistent use of names for sodium silicate and NaOH in our paper. We apologize for any confusion this may have caused and will take steps to ensure that consistent nomenclature was used throughout the paper. Thank you again for your valuable input.

(3) Others

Sodium silicate and sodium hydroxide were used as raw materials to prepare the alkali activator. The Na2O content in sodium silicate was 9.65%, the content of SiO2 was 25.22%, and the water content was 65%. Table 3 shows the specific parameters of chemical reagents. Sodium sulfide(Na2S) and glacial acetic acid were used as raw materials to prepare SEAA and heavy metal leaching agents, respectively.

Table 3. The specific parameters of chemical reagents

Chemical Reagent

Purity specification

Manufacturer

Sodium sulfide

Analytical Reagent

Fu Chen (Tianjin) Chemical Reagent Co., Ltd.

Glacial acetic acid

Analytical Reagent

Fu Chen (Tianjin) Chemical Reagent Co., Ltd.

Sodium hydroxide

Analytical Reagent

Xilong Chemical Co., Ltd.

-------------------------------------------------------------------------------------------------------

We have tried our best to revise and improve the manuscript and made the changes in the manuscript according to the Reviewer’s good comments. We appreciate for Editors/Reviewer’ warm work earnestly, and hope that the corrections will meet with approval. Once again, we acknowledge your comments and constructive suggestions very much, which are valuable in improving the quality of our manuscript.

If there are other errors or further requests, please contact us by e-mail.

Yours sincerely,

Yongsheng Ji
